# An Empirical Study Towards Prompt-Tuning for Graph Contrastive Pre-Training in Recommendations

**Haoran Yang**
University of Technology Sydney
`haoran.yang-2@student.uts.edu.au`

**Xiangyu Zhao***
City University of Hong Kong
`xianzhao@cityu.edu.hk`

**Yicong Li**
University of Technology Sydney
`yicong.li@student.uts.edu.au`

**Hongxu Chen**
University of Queensland
`hongxu.chen@uq.edu.au`

**Guandong Xu***
University of Technology Sydney
`guandong.xu@uts.edu.au`

## Abstract

Graph contrastive learning (GCL) has emerged as an effective technology for various graph learning tasks. It has been successfully applied in real-world recommender systems, where the contrastive loss and downstream recommendation objectives are combined to form the overall objective function. However, this approach deviates from the original GCL paradigm, which pre-trains graph embeddings without involving downstream training objectives. In this paper, we propose a novel framework called CPTPP, which enhances GCL-based recommender systems by leveraging prompt tuning. This framework allows us to fully exploit the advantages of the original GCL protocol. Specifically, we first summarize user profiles in graph recommender systems to automatically generate personalized user prompts. These prompts are then combined with pre-trained user embeddings for prompt tuning in downstream tasks. This helps bridge the gap between pre-training and downstream tasks. Our extensive experiments on three benchmark datasets confirm the effectiveness of CPTPP compared to state-of-the-art baselines. Additionally, a visualization experiment illustrates that user embeddings generated by CPTPP have a more uniform distribution, indicating improved modeling capability for user preferences. The implementation code is available online[2] for reproducibility.

## 1 Introduction

Graph contrastive learning (GCL) has gained significant attention in the research community as a prominent self-supervised learning paradigm. Several recent studies have showcased the effectiveness of GCL in various general graph representation tasks [21, 16, 28, 37, 30], including node classification and link prediction. Moreover, GCL has also demonstrated its applicability in real-world domains[29], such as recommender systems [27, 32, 14]. By introducing additional self-supervision signals, GCL provides recommender systems with a means to address the challenge of improving performance.

Most recommendation methods based on GCL typically combine contrastive loss with recommendation objectives to optimize the model in an end-to-end manner. However, this training protocol does

---

*Corresponding author.
[2]`https://github.com/Haoran-Young/CPTPP`

37th Conference on Neural Information Processing Systems (NeurIPS 2023).

not align with the purpose of GCL, which is primarily designed for pre-training graph representations without involving downstream task objectives [21, 16]. In this approach, GCL pre-trains embeddings that are then fine-tuned on specific tasks using downstream models. Incorporating both GCL and recommendation objectives into the overall training objective can disrupt the embedding pre-training process and requires careful control of the weight placed on contrastive loss. Additionally, previous studies on GCL-based recommendation methods [27, 14] have shown that the weights of contrastive loss in the overall objective are significantly smaller compared to the weight on the recommendation objective. This is done to ensure desired performance on recommendation tasks. Therefore, based on these observations, simply combining contrastive loss with downstream recommendation objectives may not be effective for recommendation tasks.

The disparity between the pre-training objective and downstream tasks hinders the effective extraction of useful information from pre-trained embeddings by downstream models [12, 26]. Consequently, researchers often opt for combining GCL with recommendation objectives. However, it is important to note that GCL pre-training targets primarily assess the agreement of mutual information among graph elements, such as nodes, edges, and sub-graphs. This differs from conventional graph learning tasks like node classification and link prediction. Consequently, the pre-training targets of GCL also significantly diverge from downstream recommendation objectives that involve interaction (link) prediction between users and items. Consequently, the reduction of such dissimilarities is essential to enhance the performance of GCL-based recommendation approaches.

In this paper, we present the CPTPP framework as an extension of recent advancements in prompt tuning for enhancing recommendation performance [23, 33] utilizing user embeddings pre-trained by GCL. The technique of prompt tuning has emerged as a prominent method for fine-tuning pre-trained models. By constructing appropriate prompts for downstream learning modules, this approach effectively reformulates downstream tasks, thereby reducing disparities [26, 12, 15, 18]. By incorporating prompt-tuning, we can modify existing GCL-based recommendation methods to align with the original GCL protocol involving pre-training and fine-tuning. Previous endeavors have also explored the integration of prompt learning into conventional recommendation models [26, 3]. Despite their advantages, applying the prompt mechanism directly to GCL-based recommendation methods is still difficult and not straightforward, *i.e., how can we generate personalized user prompts using only the user-item interaction graph without side information (*e.g., *age and occupation*)? To address this issue, we summarise three methods to produce different user profiles, including *historical interaction records*, *adjacency matrix factorization*, and *high-order user relations*, based on the user-item interaction graph for the personalized user prompt generation, which is applicable in situations devoid of side information. Comprehensive experiments conducted on three publicly available datasets illustrate the effectiveness of the proposed method with different types of prompts.

In summary, the contributions of this work are three-fold: (1) We propose a reformulation of existing GCL-based recommendation methods by incorporating the prompt tuning mechanism. This allows us to fully leverage the advantages of GCL during the pre-training phase, rather than relying on the combination of contrastive loss with downstream objectives. (2) We introduce three user profiles derived from the user-item interaction graph as inputs for the prompt generator. By using these profiles, we are able to generate personalized prompts that enhance the quality of user embeddings in graph-based recommendation systems. (3) We conduct extensive experiments on three publicly available benchmark datasets to validate the effectiveness of our model. Through these experiments, we analyze the important components and hyper-parameters of our approach and also investigate the impact of different personalized prompts generated by our method.

## 2 Methodology

In this section, the proposed method, graph **C**ontrastive **P**re-**T**raining with **P**rom**P**t-tuning for recommendation (**CPTPP**), will be introduced to reveal the intuitions and the technical details.

### 2.1 Framework Overview

There are three modules in the proposed CPTPP method, as shown in Figure 1: (1) graph contrastive learning module, leveraging the advantages of GCL to the pre-train user and item embeddings, (2) personalized prompts generation module, applying prompt mechanism to generate personalized

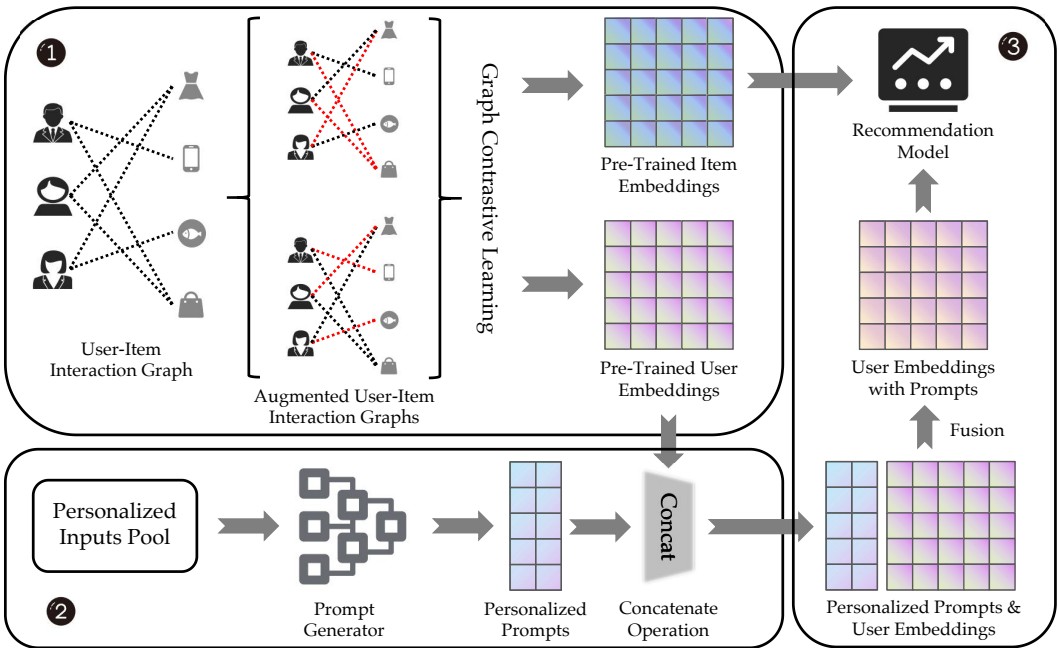

Figure 1: The overview of the proposed method CPTPP.

prompts for users, and (3) recommendation module, fusing the generated personalized prompts and pre-trained user embeddings to conduct prompt-tuning for the downstream recommendation task.

## 2.2 Graph Contrastive Learning Module

In order to achieve optimal performance in downstream tasks, the selection of a suitable pre-training strategy is crucial for generating high-quality inputs for downstream modules. GCL has been demonstrated as a powerful technique for graph pre-training [21, 16, 30, 37] and has emerged as an effective tool for leveraging self-supervised signals to enhance graph-based recommendation models [27, 14, 24, 32]. In the case of graph-based recommender systems, GCL represents a viable option for pre-training embeddings. Furthermore, our work specifically focuses on reforming and improving existing GCL-based recommendation methods. Therefore, it is imperative that we formulate a GCL module within our proposed method. Current GCL-based recommendation methods [14, 24, 27, 32] have explored various graph augmentation techniques on user-item interaction graphs in order to generate augmented graphs for GCL, enabling the extraction of informative semantics from the graph structures. Alternatively, some studies [31, 14] have designed context embeddings tailored for GCL in recommended settings. Although different approaches exist for constructing contrasting samples, they all share a common backbone for the GCL training protocol.

Here, we give a formal description of the GCL training protocol. Let $u_i$ denote the target graph element (*e.g.,* user node), $u_i^+$ represent the positive sample generated from $u_i$ (*e.g.,* the neighbor node of the target node), and $\mathcal{U}^- = \{u_{i,0}^-, u_{i,1}^-, \cdots, u_{i,n}^-\}$ be the set of contrasting samples of $u_i$ (*e.g.,* non-neighbour nodes of the target node). Considering the settings of the recommendation task, we use $G$ to represent the overall user-item graph, and all the target, positive sample, and contrasting samples are within graph $G$. To acquire embeddings of these graph elements, we adopt $f(*)$ as the graph encoder to process them, and the target embedding is denoted by $\mathbf{u}_i = f(u_i; G)$, $\mathbf{u}^+ = f(u_i^+; G)$ is the embedding of the positive sample, and $\{\mathbf{u}_{i,0}^-, \mathbf{u}_{i,1}^-, \cdots, \mathbf{u}_{i,n}^-\}$ are the list of embeddings of the negative contrasting samples. Following the settings of InfoNCE [20], the self-supervised learning objective can be formulated as follows:

$$\mathcal{L}_{contra} = -\log \frac{\exp(sim(\mathbf{u}_i, \mathbf{u}_i^+)/\tau)}{\sum_{t=0}^{|\mathcal{U}^-|} \exp(sim(\mathbf{u}_i, \mathbf{u}_{i,t}^-)/\tau)}, \tag{1}$$

where $\tau$ is the temperature hyper-parameter and $\text{sim}(\cdot, \cdot)$ is the similarity metric. In existing research works [24, 27, 32, 31, 14], researchers usually combine the aforementioned contrastive learning loss function with the recommendation objectives to formulate an overall objective function to train the model in an end-to-end manner:

$$\mathcal{L}_{overall} = \mathcal{L}_{rec} + \lambda \cdot \mathcal{L}_{contra}, \tag{2}$$

where $\lambda$ is a hyper-parameter that controls the weight of the contrastive learning objective. However, as mentioned in Section 1, the proposed CPTPP adopts a *pre-train, prompt, fine-tune* manner to train the model and treats GCL as a pre-training task instead of combining the contrastive loss with recommendation objectives. To leverage recent research progress in GCL, we can adopt various GCL learning methods tailored for the recommendation task here, like NCL [14], SGL [24], and SimGCL [32], to obtain high-quality user and item embeddings. Then, the pre-trained user and item embeddings will be processed by the prompt mechanism in the following.

## 2.3 Prompts Generation Module

Following the pre-training phase, our method, named CPTPP, incorporates a personalized prompts generation module to utilize the pre-trained user and item embeddings effectively. The primary objective of this module is to address the limitations present in existing prompt and recommendation research. Prior studies [18, 15, 26] have highlighted the triviality and resource-intensive nature of hard prompt design, making it impractical for real-world scenarios. Additionally, we observe that most current approaches rely on side information (*e.g.,* user descriptions) to generate prompts, and there lacks a specific paradigm for prompting in graph-based recommendation scenarios. To overcome these limitations, we propose the integration of a prompt generator [26] that generates personalized prompts tailored specifically for graph-based recommendation contexts.

### 2.3.1 Personalized Prompt Generator

The main scope of the generated prompts lies in narrowing the gap between the pre-training targets and the downstream objectives to utilize the pre-trained models or embeddings better. Some research works designed hard prompts tailored for recommendation tasks converting recommendation tasks into NLP tasks [3], which unifies multiple recommendation tasks in a single framework. For example, a convention recommendation task can be converted to a sentence, '*User 123 will purchase item [id_token]*'. Then, NLP techniques will be applied to predict the token. However, PPR [26] argued that such an NLP-style hard prompt designing method has two major limitations: (i) It is difficult to apply NLP techniques to predict the designated tokens since these tokens could be a user ID, item ID, or ratings, which lack meaningful semantics. (ii) The designed hard prompts are universal and cannot be customized for various users or items.

To address the challenges, we adopt a method to construct personalized prompts from user profiles in a soft prompt automatic generation manner [12, 15, 26]. Let $\mathbf{x}_i^u \in \mathbb{R}^{d \times 1}$ denote the profile of user $i$. We, then, concatenate all the users' profiles to form the user profile matrix $\mathbf{X}^u = [\mathbf{x}_1^u, \mathbf{x}_2^u, \cdots, \mathbf{x}_n^u] \in \mathbb{R}^{d \times n}$. This matrix will be fed into a two-layer perceptron $f(\cdot)$ to acquire personalized prompts for each user $\mathbf{P}^u = [\mathbf{p}_1^u, \mathbf{p}_2^u, \cdots, \mathbf{p}_n^u] \in \mathbb{R}^{p \times n}$ as follows:

$$\mathbf{P}^u = f(\mathbf{X}^u) = \mathbf{W}_2 \cdot \alpha(\mathbf{W}_1 \cdot \mathbf{X}^u + b_1) + b_2, \tag{3}$$

where $p$ is the prompt size, $\mathbf{W}_1 \in \mathbb{R}^{h \times d}$ and $\mathbf{W}_2 \in \mathbb{R}^{p \times h}$ are trainable weights, $b_1 \in \mathbb{R}^{h \times n}$ and $b_2 \in \mathbb{R}^{d \times n}$ are biases, and $\alpha(\cdot)$ is the activation function. $d$ and $h$ represent the dimensions of the pre-trained embeddings and hidden dimensions, respectively. The generated prompts will be concatenated with the pre-trained user embeddings in a pre-fixed manner [12] and tuned by the downstream objectives in the recommendation module to fulfill the process of prompt-tuning. Specifically, let $\mathbf{U}_{pre\_train} \in \mathbb{R}^{d \times n}$ denote pre-trained user embeddings. Then, we have the inputs from the user side for the recommendation module:

$$\mathbf{U}_{concat} = \begin{bmatrix} \mathbf{P}^u \\ \mathbf{U}_{pre\_train} \end{bmatrix}^T \in \mathbb{R}^{n \times (p+d)}. \tag{4}$$

### 2.3.2 Personalized Inputs for Prompt Generation

The quality of the generated prompts depends on the personalized inputs for the generator. Current research on prompt learning for recommendation mainly focuses on utilizing existing user features

(*e.g.,* age, gender, and occupation) and historical interaction records as the inputs to generate personalized prompts [26, 11, 25]. However, these methods are designed for conventional and sequential recommendations, which are not entirely aligned with graph recommendations. It is necessary to summarise and explore how to generate personalized prompts from the perspective of the graph recommendation system. In this section, we summarise three types of inputs for the generator to generate personalized prompts for the graph-based recommendation: historical interaction records, adjacency matrix factorization, and high-order user relations.

***Historical Interaction Records***. It is a common and widely-used method to illustrate users' features or preferences via aggregating his/her historical interaction records, which is feasible in various scenarios in recommendation systems. Let $\mathcal{I}_k^u = \{i_{k,1}, i_{k,2}, \cdots, i_{k,m}\}$ denote the item set which are purchased by user $k$. We use $\mathbf{i}_{k,j} \in \mathbb{R}^d$ to represent the embedding of the $j$-th item in user $k$'s purchase history. Then, the profile of user $k$ can be acquired by aggregating embeddings of those items purchased by the user $k$:

$$\mathbf{x}_k^u = Aggregation(\mathbf{i}_{k,1}, \mathbf{i}_{k,2}, \cdots, \mathbf{i}_{k,m}), \tag{5}$$

where $Aggregation(*)$ is the aggregation function to read out the user's profile.

The concatenation of all the user profiles can form the matrix $\mathbf{X}^u$ to be processed by the personalized prompt generator. Let $\mathbf{A} \in \mathbb{R}^{n \times q}$ denote the adjacency matrix for the recommendation system, which contains $n$ users and $q$ items. If we have the pre-trained item embeddings $\mathbf{I}_{pre\_train} = [\mathbf{i}_1, \mathbf{i}_1, \cdots, \mathbf{i}_q] \in \mathbb{R}^{d \times q}$, then, we have:

$$\mathbf{X}^u = (\mathbf{A} \cdot \mathbf{I}_{pre\_train}^T)^T. \tag{6}$$

***Adjacency Matrix Factorization***. The adjacency matrix is an effective tool to demonstrate the user-item relations in the recommendation system. However, the adjacency matrix usually suffers from sparsity problems and thus cannot be smoothly applied in many real-world recommendation scenarios. To address this problem, researchers proposed several matrix factorizations (MF) methods [19, 7] to decompose the adjacency matrix to obtain two matrices, $\mathbf{U}$ and $\mathbf{V}$, denoting the latent embeddings for users and items, which are much denser than the adjacency matrix $\mathbf{A}$ itself. The process of MF can be formulated as follows:

$$\arg\min_{\mathbf{U}, \mathbf{V}} \sum_{i=1}^{n} \sum_{j=1}^{q} (\mathbf{A}_{i,j} - \hat{\mathbf{A}}_{i,j}), \tag{7}$$

where $\hat{\mathbf{A}}_{i,j} = \sum_k \mathbf{U}_{i,k} \cdot \mathbf{V}_{k,j}^T = \mathbf{U}_i \mathbf{V}_j^T$. After the MF process, we have the latent matrix of users, $\mathbf{U} \in \mathbb{R}^{n \times d}$, serving as the user profile matrix $\mathbf{X}^u$ after transposed $\mathbf{X}^u = \mathbf{U}^T$ and can be fed into a personalized prompt generator to produce personalized prompts $\mathbf{P}^u$ for each user. Specifically, we set the size of latent embeddings as $d$, the same as the size of pre-trained embeddings.

***High-Order User Relations***. Learning informative embeddings from a 1-hop user-item interaction graph is challenging when there is no side information. To address this limitation, we propose to leverage high-order user relations to enrich the learned embeddings via constructing 2-ego graphs for each user node to find the links between the other users and itself [22]. Then, we fuse the target user's purchase history and high-order neighbor embeddings to represent the target user profile.

We first construct the high-order connectivity matrix to achieve the goal. Let $\mathbf{A}^* = \bar{\mathbf{A}} \cdot \bar{\mathbf{A}} \in \mathbb{R}^{(n+q) \times (n+q)}$ denote the high-order connectivity matrix, where $\bar{\mathbf{A}} = \begin{bmatrix} \mathbf{0} & \mathbf{A} \\ \mathbf{A}^T & \mathbf{0} \end{bmatrix} \in \mathbb{R}^{(n+q) \times (n+q)}$, recording all the users and items to which a user or item node is connected. Then, we build the matrix $\mathbf{E} = [\mathbf{U}_{pre\_train}, \mathbf{I}_{pre\_train}]^T \in \mathbb{R}^{(n+q) \times d}$ to store pre-trained embeddings. Next, we can acquire matrix $\mathbf{Q} \in \mathbb{R}^{n \times d}$, which are the users' personalized profiles about high-order user relations, via:

$$\begin{bmatrix} \mathbf{Q} \\ \mathbf{M} \end{bmatrix} = \mathbf{A}^* \cdot \mathbf{E} \in \mathbb{R}^{(n+q) \times d}, \tag{8}$$

where $\mathbf{M} \in \mathbb{R}^{q \times d}$, denoting the high-order item relations and being omited after $\mathbf{Q}$ is extracted. Then, a matrix transpose operation is required to obtain the user profile matrix $\mathbf{X}^u = \mathbf{Q}^T$.

## 2.4 Recommendation Module

After the pre-training and the personalized prompts generation phase, a recommendation module is required so that we can verify whether the prompt-tuning module rectifies the pre-trained embeddings by GCL and makes them be adapted to the downstream recommendation tasks better. In this module, we take the inner product of user and item embeddings as the predicted score for the recommendation. Bayesian Personalized Ranking (BPR) [17] is adopted as the training objective to tune the pre-trained embeddings based on the predicted scores. The motivation for formulating such a simple recommendation module is to avoid the performance gain brought by the delicate designs of those advanced recommendation models, which could affect the observations on our proposed method.

---

**Algorithm 1:** CPTPP algorithm

---

**Input:** User embedding table $\mathbf{U}_E$; Item embedding table $\mathbf{I}_E$; User-item interaction graph adjacency matrix $\mathbf{A}$; Graph contrastive learning model $f(*)$; User profile $\mathbf{X}^u$; Prompt generator $g(\cdot)$; Multi-layer perceptron MLP $(\cdot)$; Pre-train epoch $i$; Prompt-tune epoch $j$.

**Output:** User and item embedding tables $\mathbf{U}_E^*$ and $\mathbf{I}_E^*$.

1   *Pre-train phase:*
2   *Initialize $\mathbf{U}_E, \mathbf{I}_E$; $\mathbf{U}_E^{'}, \mathbf{I}_E^{'} \leftarrow \mathbf{U}_E, \mathbf{I}_E$;*
3   $count = 0$;
4   **while** $count < i$ **do**
     // Update user and item embedding tables.
5      $\mathbf{U}_E^{'}, \mathbf{I}_E^{'} = f(\mathbf{U}_E^{'}; \mathbf{I}_E^{'}; \mathbf{A})$;
6      $count = count + 1$;
7   **end**
8   *Prompt-tune phase:*
9   $\mathbf{U}_E^* \leftarrow \mathbf{U}_E^{'}$; $\mathbf{I}_E^* \leftarrow \mathbf{I}_E^{'}$;
10   $count = 0$;
11   **while** $count < j$ **do**
     // Personalized prompt generation.
12      $\mathbf{P}^u = g(\mathbf{X}^u)$;
     // Concatenate & fusion.
13      $\mathbf{U}_E^* = \mathtt{MLP}([\mathbf{P}^u; \mathbf{U}_E^*]^T) \in \mathbb{R}^{n \times d}$;
14      *Optimise $\mathcal{L} = \sum_{i \in \mathcal{U}} \mathcal{L}_{rec}^i + \lambda ||\Theta||_2^2$;*
15      *Update $\mathbf{U}_E^*, \mathbf{I}_E^*$;*
16      $count = count + 1$;
17   **end**
18   **return** $\mathbf{U}_E^*$, $\mathbf{I}_E^*$

---

### 2.4.1 Prompts and Pre-Trained Embeddings Fusion

We concatenate the generated personalized prompts and the pre-trained user embeddings in the previous step and have $\mathbf{U}_{concat} \in \mathbb{R}^{n \times (p+d)}$, whose dimensionality is different from the pre-trained embeddings $\mathbf{I}_{pre\_train} \in \mathbb{R}^{n \times d}$. Hence, we need first to fuse the personalized prompts and the pre-trained user embeddings for the recommendation objective training. Specifically, we adopt a multi-layer perceptron (MLP) $g(\cdot)$ as the mapping function that is $g : \mathbb{R}^{n \times (p+d)} \to \mathbb{R}^{n \times d}$. Then, we can have dimensionality-reduced user representations $\mathbf{U}^* = g(\mathbf{U}_{concat}) \in \mathbb{R}^{n \times d}$, enhanced by the personalized prompts. After that, we can apply the inner product to predict how likely the user $i$ would interact with the item $j$ by $\hat{y}_{i,j} = \mathbf{u}_i^* \cdot \mathbf{i}_j$, where $\mathbf{u}_i^*$ is the $i$-th row of $\mathbf{U}^*$.

### 2.4.2 Training Objective for Recommendation Task

For simplicity and fair comparison, we adopt BPR [17] loss as the training objective for the recommendation task. For each user $i$, we have:

$$\mathcal{L}_{rec}^i = \sum_{j^+ \in \mathcal{I}_i^u} \sum_{j^- \in \mathcal{I} \backslash \mathcal{I}_i^u} -\log \sigma(\hat{y}_{i,j^+} - \hat{y}_{i,j^-}). \tag{9}$$

However, it is unaffordable to consider all the unobserved interactions of the user $i$. Therefore we sample several negative items $\mathcal{N}_i^u$, where $|\mathcal{N}_i^u| << |\mathcal{I}\backslash\mathcal{I}_i^u|$, in practice.

Moreover, we introduce $L2$-norm into the training objective to regularize the parameters in the model to address the overfitting problem and improve generalization ability. Therefore, the overall objective function can be formulated as:

$$\mathcal{L} = \sum_{i \in \mathcal{U}} \mathcal{L}_{rec}^i + \lambda ||\Theta||_2^2. \tag{10}$$

## 2.5 Summary

After the training process ends, the model can be used to conduct inference. For the inference phase, we do not conduct pre-train and prompt-tune again. What we need to do is to extract target user and item embeddings from the trained embedding tables. Then, we calculate their inner product to predict the probability that the user will interact with the item in the future.

The complete training procedure of CPTPP is illustrated by Algorithm 1. We first initialize the user and item embedding tables (line 2). Then, we apply a GCL model to conducting embedding pre-training (line 4 ~ 7). Next, we step into the prompt-tuning phase and assign the pre-trained embeddings to $\mathbf{U}_E^*$ and $\mathbf{I}_E^*$ (line 9). Following, we input the user profile to the prompt generator to produce the personalized prompts (line 12) and combine them with $\mathbf{U}_E^*$ (line 13). Finally, we use $\mathbf{U}_E^*$ and $\mathbf{I}_E^*$ to calculate the loss and update them accordingly (line 14 ~ 15). The update procedure will repeat until the termination condition is achieved (line 11 ~ 17).

# 3 Experiment

To verify the effectiveness of the proposed method, CPTPP, in this paper, we conduct extensive experiments and demonstrate the results with insightful analysis in this section.

Table 1: Dataset Statistics

| Dataset | #Users | #Items | #Interactions | Density |
|---------|--------|--------|---------------|---------|
| Douban | 2,848 | 39,586 | 894,887 | 0.794% |
| ML-1M | 6,040 | 3,900 | 1,000,209 | 4.246% |
| Gowalla | 29,858 | 40,981 | 1,027,370 | 0.084% |

## 3.1 Experimental Setup

This section introduces the experimental settings, including datasets and baselines we used, performance metrics, and hyper-parameter settings for CPTPP. More details about how to get access to the datasets and implementation details are listed in **Appendix A**.

**Datasets** To verify the performance of CPTPP in the recommendation task, we select three popular datasets: **Douban** [34], **MovieLens-1M** [5], and **Gowalla** [13]. The detailed statistics about the three datasets are listed in Table 1. For each dataset, we randomly select 80% of historical user-item interactions as the training set, and the rest 20% records will serve as the testing set. Following the settings widely adopted by the research community [22, 6], we treat each user-item interaction record as a positive instance and conduct negative sampling to couple it with a negative instance, which is an unobserved user-item interaction in the dataset.

**Baselines** We select several baselines for comparison experiments: **BPR-MF** [9], **BUIR** [10], **SelfCF** [36], **NCL** [14], and **SimGCL** [32]. For CPTPP, we have three variations, which are CPTPP-H, CPTPP-M, and CPTPP-R, respectively. *-H* takes historical interaction records for personalized prompt generation. *-M* indicates that we take adjacency matrix factorization for personalized prompts generation. Furthermore, *-R* takes high-order user relations for the personalized prompt generation.

**Metrics** To evaluate the quality of top-$K$ recommendation, we adopt three popular metrics, which are *Hit Ratio@K*, *Precision@K*, and *NDCG@K*, respectively. In our settings, the value of $K$ is set to 5 and 20. Following the evaluation protocol in [14, 32], we take the full ranking strategy [35].

**Hyper-parameter Settings** To ensure reproducibility, we disclose the comprehensive hyper-parameter settings for implementing our proposed CPTPP in the source codes and **Appendix A.3**.

## 3.2 Experiment Results

We conduct experiments and provide analysis in this section. More supplementary experiment results, including comparison, hyper-parameter, and ablation study, are revealed in **Appendix B**.

Table 2: The experiment results of comparison studies. The figures in boldface indicate the best performance achieved by one of the three versions of CPTPP, and the figures underlined indicate the best performance among all the baselines.

| Datasets | Metrics | Methods | | | | | | | |
|---|---|---|---|---|---|---|---|---|---|
| | | BPR-MF | BUIR | SelfCF | NCL | SimGCL | CPTPP-H | CPTPP-M | CPTPP-R |
| Douban | Hit Ratio@5 | 0.0134 | 0.0156 | 0.0161 | 0.0161 | 0.0161 | 0.0164 | **0.0165**\* | 0.0164 |
| | Hit Ratio@20 | 0.0446 | 0.0492 | 0.0502 | 0.0507 | 0.0489 | 0.0521 | **0.0528**\* | 0.0523 |
| | Precision@5 | 0.1812 | 0.2113 | 0.2185 | 0.2187 | 0.2182 | 0.2221 | **0.2235**\* | 0.2224 |
| | Precision@20 | 0.1512 | 0.1667 | 0.1699 | 0.1717 | 0.1657 | 0.1766 | **0.1790**\* | 0.1772 |
| | NDCG@5 | 0.1904 | 0.2209 | 0.2264 | 0.2313 | 0.2370 | 0.2359 | **0.2378**\* | 0.2355 |
| | NDCG@20 | 0.1749 | 0.2019 | 0.2058 | 0.1958 | 0.2020 | 0.2065 | **0.2098**\* | 0.2070 |
| ML-1M | Hit Ratio@5 | 0.0469 | 0.0617 | 0.0624 | 0.0655 | 0.0631 | **0.0676**\* | 0.0674 | 0.0672 |
| | Hit Ratio@20 | 0.1454 | 0.1519 | 0.1643 | 0.1796 | 0.1698 | 0.1851 | **0.1861**\* | 0.1845 |
| | Precision@5 | 0.1800 | 0.2368 | 0.2396 | 0.2513 | 0.2420 | **0.2592**\* | 0.2585 | 0.2577 |
| | Precision@20 | 0.1395 | 0.1457 | 0.1576 | 0.1723 | 0.1629 | 0.1776 | **0.1785**\* | 0.1770 |
| | NDCG@5 | 0.1968 | 0.2722 | 0.2689 | 0.2818 | 0.2767 | **0.2919**\* | 0.2895 | 0.2878 |
| | NDCG@20 | 0.2103 | 0.2367 | 0.2508 | 0.2683 | 0.2670 | 0.2781 | **0.2782**\* | 0.2756 |
| Gowalla | Hit Ratio@5 | 0.0429 | 0.0479 | 0.0497 | 0.0488 | 0.0513 | 0.0518 | 0.0512 | **0.0519**\* |
| | Hit Ratio@20 | 0.1039 | 0.0993 | 0.1042 | 0.1040 | 0.1065 | 0.1115 | 0.1103 | **0.1120**\* |
| | Precision@5 | 0.0624 | 0.0698 | 0.0723 | 0.0711 | 0.0746 | 0.0754 | 0.0745 | **0.0755**\* |
| | Precision@20 | 0.0378 | 0.0361 | 0.0379 | 0.0378 | 0.0387 | 0.0406 | 0.0401 | **0.0407**\* |
| | NDCG@5 | 0.0770 | 0.0911 | 0.0939 | 0.0894 | 0.0963 | **0.0963** | 0.0953 | 0.0961 |
| | NDCG@20 | 0.0939 | 0.0990 | 0.1036 | 0.1005 | 0.1126 | **0.1092** | 0.1083 | **0.1092** |

"**\***" indicates that CPTPP outperforms the best baseline significantly (i.e., two-sided t-test with $p < 0.05$).

### 3.2.1 Overall Comparison Studies

Table 2 shows the comparison results among all the baselines and different versions of the proposed methods. (i) We can first observe that the traditional method BPR-MF is outperformed by all the other methods as they utilize contrastive learning to introduce extra unsupervised training signals. (ii) Among all the baselines, GCL-based recommendation methods, including NCL and SimGCL, significantly and consistently outperform those self-supervised recommendation methods without graph learning module equipped, BUIR and SelfCF. It is because those GCL-based methods adopt graph neural networks, leveraging the sophisticated structure semantics in user-item interaction graphs to enrich learned user embeddings and item embeddings. (iii) But we notice that SimGCL only outperforms NCL on dataset Gowalla, which has a much larger scale than the others, probably because SimGCL adopts a simplified GCL method that relieves the model overfitting problem on a large-scale dataset. It is the potential reason NCL outperforms SimGCL on smaller datasets, as the simplified GCL method may not provide sufficient self-supervised training signals. (iv) Though the proposed CPTPP solely adopts BPR loss, which is significantly different from the pre-training procedure, for the recommendation task training, we utilize the prompt learning mechanism to better adapt the embeddings pre-trained by the GCL method to the downstream task, expecting to improve the recommendation performance. According to the experiment results, all versions of our proposed method achieve competitive results. Such results reflect prompt-tuning's effectiveness in narrowing the gap between the pre-train objective and the downstream tasks.

To further evaluate the performance of the GCL-based recommendation methods, we visualize the produced user embeddings produced by t-SNE and Gaussian kernel density estimation (KDE). We can see that CPTPP has a more uniform distribution of the produced user embeddings, illustrated by the uniformity of the color maps, especially on dataset ML-1M and Gowalla. As suggested by Z. Lin *et al.* [14], the more uniform the embedding distribution is, the more powerful the capability to model the diverse preferences of users the produced embeddings will have, which reflects the superiority of CPTPP compared to the baselines. The visualizations and analysis are listed in **Appendix B.1**.

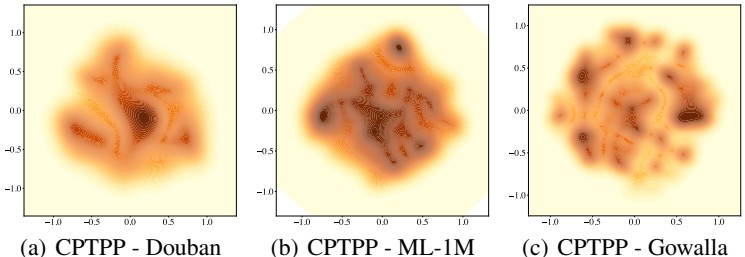

(a) CPTPP - Douban     (b) CPTPP - ML-1M     (c) CPTPP - Gowalla

Figure 2: The visualizations of the user embeddings generated by the proposed method.

### 3.2.2 Hyper-Parameter Studies

To investigate the properties of our proposed CPTPP method, we conduct hyper-parameter studies on an important term, the dimension size of the personalized prompt. By fixing all the other hyper-parameters, we comprehensively examine the performance of three versions of the proposed CPTPP on all the datasets with different prompt sizes. Specifically, the size of the personalized prompt is selected from $\{8, 16, 32, 64, 128, 256\}$. We choose two metrics, *Precision@5* and *NDCG@5*, to demonstrate CPTPP's performance variations with regard to different prompt sizes. All the experiment results are shown in Figure 3 and Figure 5 in **Appendix B.2**. (i) The first finding we can observe is that, in most cases, CPTPP has the best performance when the prompt size is not larger than the dimensionality of user embeddings, *i.e.,* 64. A potential reason is that the prompt is usually less informative than the pre-trained embeddings, so a sizeable prompt dimension would introduce too much noise to disturb and conceal the structural semantics contained in the pre-trained user embeddings. (ii) We also notice a significant performance improvement when prompt size is 256 in several cases, such as CPTPP-M on dataset ML-1M and CPTPP-R on dataset Gowalla. Such outlier performance could be caused by random factors during the overall training process. However, they still fail to significantly outperform the CPTPP model, which has a much smaller prompt size. Therefore, a small prompt size for prompt-tuning is a better option in practice as they achieve a relatively better recommendation quality and higher efficiency.

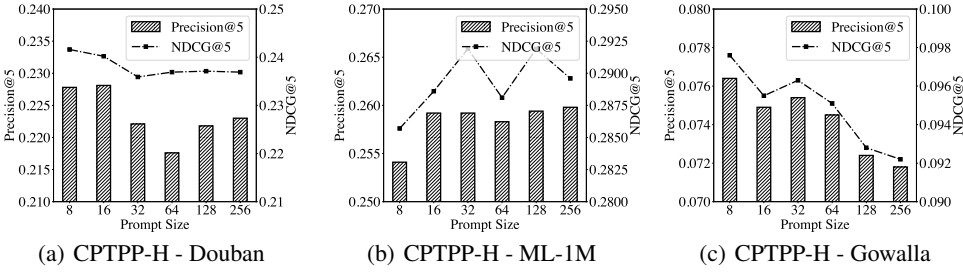

(a) CPTPP-H - Douban     (b) CPTPP-H - ML-1M     (c) CPTPP-H - Gowalla

Figure 3: The performance, demonstrated by *Precision@5* and *NDCG@5*, of all variations of CPTPP.

### 3.2.3 Ablation Studies

As we summarize three strategies to generate personalized prompts for users, we conduct the ablation study to explore the differences among these methods. Two ablation studies are conducted in this section to illustrate the performance of three variations of the CPTPP method. The first ablation study is about the overall evaluation of recommendation quality, whose analysis is listed below. The second one is about the embedding visualizations and the related analysis, illustrated in **Appendix B.3**.

We notice that (i) CPTPP-M achieves the best performance on dataset Douban. Nevertheless, the performance of CPTPP-M degrades on dataset ML-1M and is the worst case on dataset Gowalla. Considering the number of users reflected in Table 1, we find that the performance of CPTPP-M drops as the dataset's number of users increases. So, CPTPP-M has good performance if the number of users in the dataset is relatively small. It may be because matrix factorization, as a naive method, cannot

fully reveal user preferences in a complex user-item interaction graph with too many user nodes. (ii) CPTPP-R utilizes high-order relationships among users to enrich the generated personalized prompts for users. In such settings, the item information would also be aggregated due to the message-passing mechanism in GNNs. Therefore, it achieves the best performance on the dataset Gowalla, having the most users and the most complex user-user relation among all the datasets. (iii) CPTPP-H has moderate performance. CPTPP-H adopts historical interaction records, formed by trainable item embeddings, to generate personalized prompts. Those trainable elements endow CPTPP-H with a more robust capability to represent user preferences than matrix factorization. It is also reasonable that CPTPP-R outperforms CPTPP-H as CPTPP-H lacks consideration of high-order user relations.

## 4 Related Work

### 4.1 GCL in Recommendation Systems

The GCL-based recommendation system is now a trending topic in the research community. Researchers leverage the advantages of GCL to improve current graph-based recommendation methods further, achieving satisfying performance. The research scope first lies in utilizing user-item interaction records to accommodate GCL techniques. For example, HMG-CR [27] innovatively proposes a concept called hyper-meta path. Then GCL is applied to adaptively learn the behavioural patterns of users via contrasting different paths. NCL [14] improves graph-based recommendation models via neighborhood-enriched contrastive learning, including semantic neighbors and structural neighbors ($r$-ego graphs). To better understand the role of GCL in the recommendation systems, SGL [24] conducts a comprehensive theoretical analysis, giving insights and achieving promising performance. Based on the findings of SGL, SimGCL [32] is proposed to simplify GCL in recommendation via discarding complex augmentations, reducing the volume of GCL-based recommendation models, and performing competitive results in the experiments. However, current GCL-based recommendation methods combine the graph contrastive loss with the recommendation objectives to formulate an overall objective to train the models, suffering from various limitations summarised in the previous sections. Our method innovatively introduces the prompt mechanism to build a *pre-train then prompt-tune* paradigm for GCL-based recommender systems to address the limitations we discussed.

### 4.2 Prompt-Tuning

Prompt-tuning is a novel and trending paradigm for pre-train models in the natural language process (NLP). The core idea of prompt-tuning is to re-formulate the downstream tasks, narrowing the huge gap between them and the pre-train objective [1, 18]. There are two types of methods to achieve prompt-tuning [3]. The first is manually designing or searching for proper discrete prompts (hard prompts) [2, 8, 18]. However, such a fashion is trivial and resource-consuming as the search space is extremely large and expert knowledge is required [26] in some application scenarios. To address this limitation, another line of methods is proposed, focusing on generating continuous vector embeddings as the soft prompts [4, 15]. The application of prompt-tuning in recommendation systems has been explored. For example, P5 [3] reforms the recommendation tasks to the NLP tasks and follows the hard-prompt fashion to perform recommendations. PPR [26], instead, takes a soft-prompt and pre-fix strategy [12] to generate personalized prompts for users in the recommendation systems automatically. Nevertheless, graph learning and its applications are now out of the scope of prompt-tuning research. Besides, most current prompt learning methods require side information to produce high-quality prompts, resulting in a limited comfort zone for its applications. Our CPTPP first adopts the prompt mechanism to the GCL-based recommendation area.

## 5 Conclusion

In this paper, we propose a CPTPP method to adopt a prompt-tuning technique to reform and improve current GCL-based recommendation methods. To better accommodate prompt learning to graph recommendation scenarios, we summarise several graph-oriented user profiles to generate personalized user prompts to conduct prompt-tuning for downstream recommendation tasks. Comprehensive experiments have shown the effectiveness, superiority, and properties of the proposed CPTPP method. The future research directions about prompt-tuning in GCL-based recommendation may be two-fold: how to (i) generate personalized prompts and (ii) integrate prompt-tuning strategy into GCL protocols.

## Acknowledgments and Disclosure of Funding

This research work was supported by the Research Impact Fund (No. R1015-23), the Australian Research Council (ARC) under Grant Nos. DP220103717, LE220100078, LP170100891, and DP200101374 and was partially supported by APRC - CityU New Research Initiatives (No. 9610565, Start-up Grant for New Faculty of City University of Hong Kong), CityU - HKIDS Early Career Research Grant (No. 9360163), Hong Kong ITC Innovation and Technology Fund Midstream Research Programme for Universities Project (No. ITS/034/22MS), Hong Kong Environmental and Conservation Fund (No. 88/2022), SIRG - CityU Strategic Interdisciplinary Research Grant (No.7020046, No. 7020074), Tencent (CCF-Tencent Open Fund, Tencent Rhino-Bird Focused Research Fund), Huawei (Huawei Innovation Research Program), Ant Group (CCF-Ant Research Fund, Ant Group Research Fund) and Kuaishou.

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

# A  Reproducibility

This section provides supplementary details about our experimental settings for reproducibility.

## A.1  Datasets

Three publicly available datasets are used in this work to examine the performance of the proposed CPTPP. Here, we provide the links for downloading these datasets for readers to retrieve:

- **Douban**: `https://pan.baidu.com/s/1hrJP6rq#list/path=%2F`

- **ML-1M**: `https://grouplens.org/datasets/movielens/1m/`

- **Gowalla**: `https://github.com/kuandeng/LightGCN/tree/master/Data/gowalla`

## A.2  Baselines

We select several baselines for comparison experiments, which are listed below:

- **BPR-MF** [9] adopts a matrix factorization framework to learn embeddings for users and items via optimizing the BPR loss function.

- **BUIR** [10] only uses positive user-item interactions to learn representations following a bootstrapped manner, consisting of an online encoder and a target encoder.

- **SelfCF** [36] follows a similar strategy that BUIR adopts, which drops the momentum encoder to simplify the previously proposed method.

- **NCL** [14] utilizes neighbor clustering to enhance GCL methods to acquire enhanced embeddings for users and items in the recommendation system.

- **SimGCL** [32] discusses the role of augmentations in GCL for recommendation tasks and proposes a simplified GCL method for recommendations.

## A.3  Hyper-parameter Settings

Table 3: Summary of hyper-parameter settings of CPTPP.

| Hyper-parameter | Notation | Dataset | | |
|---|---|---|---|---|
| | | Douban | ML-1M | Gowalla |
| Hidden dimension size | $d$ | 64 | 64 | 64 |
| Pre-train epoch | - | 10 | 10 | 10 |
| Prompt-tune epoch | - | 100 | 100 | 100 |
| Batch size | - | 512 | 512 | 2048 |
| Learning rate | - | 0.003 | 0.001 | 0.001 |
| Regularizer weight | $\lambda$ | 0.0001 | 0.0001 | 0.0001 |
| Number of GNN layers | - | 2 | 2 | 2 |
| Dropout rate | - | 0.1 | 0.1 | 0.1 |
| Temperature parameter | $\tau$ | 0.2 | 0.2 | 0.2 |
| Prompt size | $p$ | {8, 16, 32, 64, 128, 256} | {8, 16, 32, 64, 128, 256} | {8, 16, 32, 64, 128, 256} |

We list detailed hyper-parameter settings here for reproducibility. The dimensionality of the representation embeddings of users and items is set to 64, and the personalized prompt size is chosen from $\{8, 16, 32, 64, 128\}$. For the pre-train phase, the maximum training epoch is 10, and for the prompt-tune stage, the training epoch is set to 100. The training batch size is 512 for the relatively smaller datasets, including Douban and ML-1M. For Gowalla, it is set to 2048. The learning rate and $\lambda$ are set to $1e^{-3}$ and $1e^{-4}$, where $\lambda$ is the weight for the $l2$-norm term in the overall training objective. The default number of layers of graph neural networks used in the models is set to 2. These settings are summarised in Table 3.

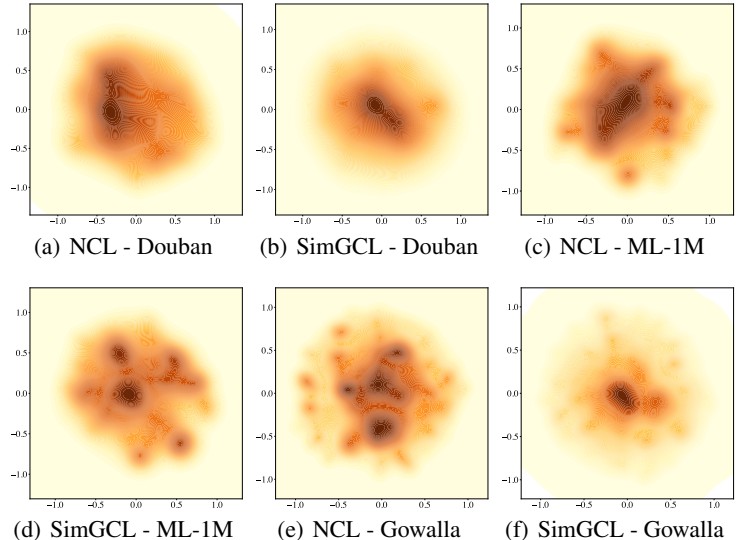

(a) NCL - Douban     (b) SimGCL - Douban     (c) NCL - ML-1M

(d) SimGCL - ML-1M     (e) NCL - Gowalla     (f) SimGCL - Gowalla

Figure 4: The visualization results of the user embeddings generated by baselines.

# B   Supplementary Experiment

In this section, several supplementary experiments are provided. Due to the page limit, the supplementary experiment results are listed and analyzed in the following instead of the main content.

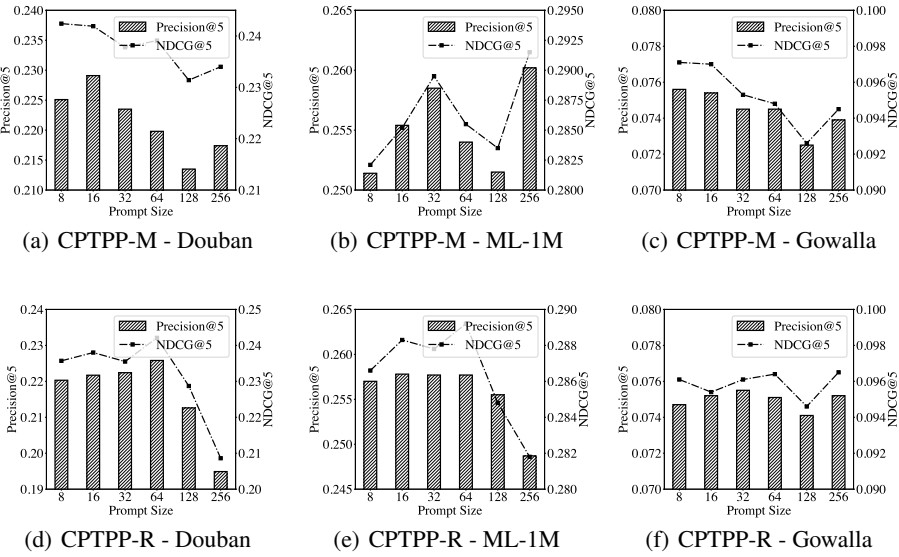

(a) CPTPP-M - Douban     (b) CPTPP-M - ML-1M     (c) CPTPP-M - Gowalla

(d) CPTPP-R - Douban     (e) CPTPP-R - ML-1M     (f) CPTPP-R - Gowalla

Figure 5: The performance, demonstrated by the metrics *Precision*@5 and *NDCG*@5, of CPTPP-M and CPTPP-R on the selected datasets.

## B.1   Supplementary Comparison Study

According to Figure 4, We can see that (i) embeddings learned by SimGCL fall into several hot areas on dataset ML-1M, and they are centralized in a small area on datasets Douban and Gowalla. (ii) NCL exhibits better performance as the distribution of the user embeddings expands to a relatively larger area than that of SimGCL. Compared to our proposed method CPTPP, we can observe that CPTPP has a more uniform distribution of the produced user embeddings, illustrated by the uniformity of the

color maps, especially on dataset ML-1M and Gowalla. As suggested in [14], the more uniform the embedding distribution is, the more capability to model the diverse preferences of users the method has, which reflects CPTPP's superiority.

## B.2 Supplementary Hyper-Parameter Study

In Section 3.2.2, we solely illustrate the performance of CPTPP-H with different prompt sizes on the three datasets. We show the rest of the hyper-parameter study results in Figure 5. For easy reading, we list our findings here again: (i) The first thing we can observe is that, in most cases, CPTPP has the best performance when the prompt size is not larger than the dimensionality of user embeddings. A potential reason is that sizeable prompt dimensions would introduce more noise into pre-trained user embeddings, disturbing the structural semantics extracted from the user-item interaction graph by graph contrastive learning. (ii) We also notice a significant performance improvement when prompt size is 256 in several cases, such as CPTPP-M on dataset ML-1M and CPTPP-R on dataset Gowalla. However, they still fail to significantly outperform the CPTPP model, which has a much smaller prompt size. Therefore, a small prompt size for prompt-tuning is a better option in practice as they achieve a relatively good recommendation quality and higher efficiency.

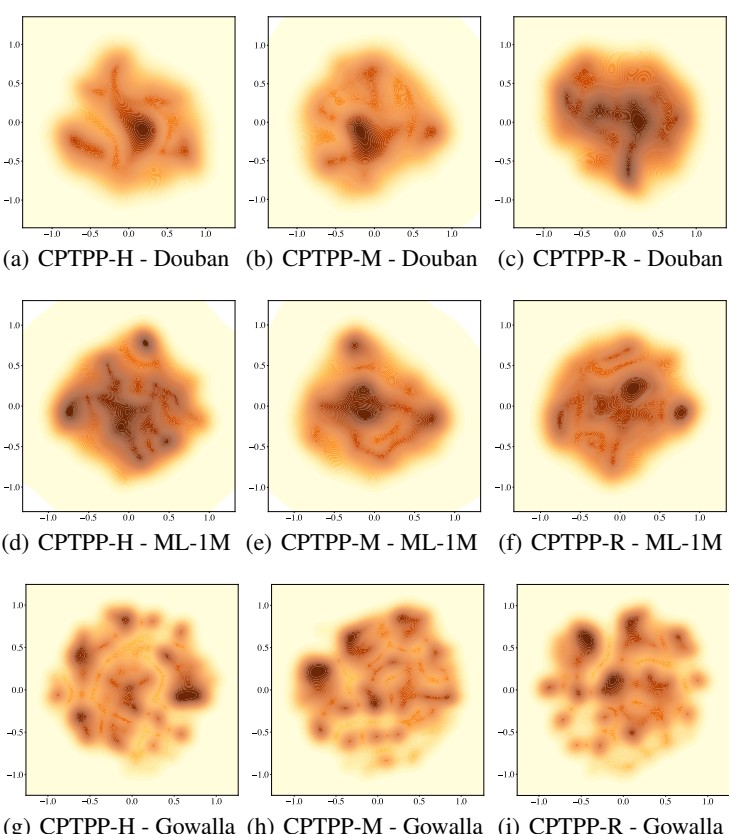

(a) CPTPP-H - Douban  (b) CPTPP-M - Douban  (c) CPTPP-R - Douban

(d) CPTPP-H - ML-1M  (e) CPTPP-M - ML-1M  (f) CPTPP-R - ML-1M

(g) CPTPP-H - Gowalla  (h) CPTPP-M - Gowalla  (i) CPTPP-R - Gowalla

Figure 6: The visualizations of the user embeddings generated by different versions of CPTPP.

## B.3 Supplementary Ablation study

The impacts of different personalized prompts on CPTPP are investigated. We visualize the user embeddings produced by all three variations of the proposed CPTPP as shown in Figure 6. We can observe that both CPTPP-H and CPTPP-R have a more uniform distribution, especially on datasets Douban and ML-1M. Such an observation indicates that personalized prompts generated from trainable user profiles can produce user embeddings that have more uniform distributions to demonstrate diverse user preferences better.

III