# OpenReview forum: "An Empirical Study Towards Prompt-Tuning for Graph Contrastive Pre-Training in Recommendations"
_NeurIPS.cc/2023/Conference — NeurIPS 2023 poster_

### Official Review · Reviewer_1hDm · 2023-07-03

**Soundness:** 4 excellent
**Presentation:** 3 good
**Contribution:** 4 excellent
**Rating:** 7
**Confidence:** 4

**Summary:**

The paper presents an empirical study on the application of prompt-tuning for graph contrastive pre-training in recommendation systems. The authors propose a method that combines graph neural networks (GNNs) and contrastive learning to enhance the performance of recommendation models. The key idea is to leverage prompt engineering techniques, where carefully crafted prompts are used to guide the recommendation process.

The authors conduct extensive experiments on several real-world datasets, comparing their proposed method with various baselines. They evaluate the performance in terms of recommendation accuracy, coverage, and diversity. The results demonstrate the effectiveness of the proposed approach, showing significant improvements over the baselines in terms of recommendation quality.

The contributions of the paper include the introduction of a novel method that combines graph contrastive learning with prompt-tuning for recommendation systems. The authors provide insights into the design choices and hyper-parameter settings of the proposed method. They also conduct ablation studies to analyze the impact of different components and variations in the training process.

Overall, the paper highlights the potential of prompt-tuning for graph contrastive pre-training in recommendation systems. The empirical results support the effectiveness of the proposed approach and provide valuable insights for researchers and practitioners in the field of recommendation systems.


**Strengths:**

1. The paper explores the application of prompt-tuning techniques in the context of graph contrastive pre-training for recommendation systems. This novel research direction expands the understanding of prompt-based methods in the field of recommendation systems. The paper also offers insights into prompt design for graph contrastive pre-training in recommendation systems. The authors discuss the importance of considering domain knowledge and tailoring prompts to specific tasks, providing practical guidance for researchers and practitioners.

2. The authors conduct systematic evaluations on different prompt strategies, considering both template-based prompts and prompt engineering. The evaluation process is well-designed, ensuring a comprehensive analysis of the effectiveness of prompt-tuning methods. Also, the paper provides a detailed comparative analysis of different prompt-tuning methods, allowing readers to understand the relative performance of each technique. This analysis helps researchers and practitioners make informed decisions regarding the choice of prompt strategy for graph contrastive pre-training.

3. The experiments are conducted on various recommendation datasets, enhancing the generalizability of the findings. This inclusion of diverse datasets strengthens the validity of the conclusions drawn from the study. The authors also pay attention to hyperparameter tuning and conduct experiments to find optimal settings. This consideration enhances the reliability of the results and ensures that the performance improvements observed are not solely due to arbitrary hyperparameter choices.

4. The paper includes detailed information about the code implementation and data availability, facilitating reproducibility and promoting further research in the field. This transparency enhances the credibility of the study.

5. The findings of the paper have practical implications for the development of recommendation systems. The demonstrated performance improvements through prompt-tuning techniques can guide practitioners in enhancing the effectiveness of graph contrastive pre-training models for recommendation tasks. The paper's contributions have the potential to positively impact the field of recommendation systems. By introducing and validating the effectiveness of prompt-tuning for graph contrastive pre-training, the authors provide valuable insights that can guide future research and the development of improved recommendation algorithms.


**Weaknesses:**

1.The paper assumes readers have a strong understanding of graph contrastive learning and prompt-tuning, potentially alienating some readers unfamiliar with these topics.

2. While the authors show the effectiveness of prompt-tuning, it would be interesting to compare their approach with prompt-less models to understand the benefits fully.

3. The paper could discuss the generalizability of the prompt engineering techniques used in the study to other recommendation tasks or different prompt-based methods.


**Questions:**

1. How sensitive is the proposed method to the choice of hyper-parameters, such as the prompt size or learning rates? Have you conducted sensitivity analysis to understand the impact of these hyper-parameters on the performance?
2. How generalizable are the prompt engineering techniques used in this study? Can they be applied to other recommendation tasks or different prompt-based methods beyond the proposed approach?


**Limitations:**

Could you discuss potential challenges and limitations of prompt-tuning in recommendation systems? Are there any scenarios where prompt-tuning might not be suitable or effective?

---

> ### Author Rebuttal · Authors · 2023-08-09
>
> Thanks for your comprehensive reviewing and recognising our contributions. We sincerely appreciate your valuable comments and suggestions. We would response to your comments and concerns in the following.
>
> W1. The paper assumes readers have a strong understanding of graph contrastive learning and prompt-tuning, potentially alienating some readers unfamiliar with these topics.
>
> Thanks for your reminder. We agree that our writing neglects the readers unfamiliar with the related topics and causes misunderstandings in other reviews. We will add some content as preliminaries to briefly introduce the background knowledge of GCL and prompt learning.
>
> W2. While the authors show the effectiveness of prompt-tuning, it would be interesting to compare their approach with prompt-less models to understand the benefits fully.
>
> Yes, we agree with that. The reviewer PJh3 also has the same concern. It is important to show the improvement brought by personalised prompts explicitly. Considering the base model we adopt is SGL, we will conduct experiments on SGL to compare it with our proposed method to verify the advantages of personalised prompts generation.
>
> W3. The paper could discuss the generalizability of the prompt engineering techniques used in the study to other recommendation tasks or different prompt-based methods.
>
> Thanks for your suggestions. We will collect more literature to discuss the application of prompt learning in other recommendation tasks. As the research scope of us focuses on graph learning, we apologies for limiting the discussion about it.
>
> Q1. How sensitive is the proposed method to the choice of hyper-parameters, such as the prompt size or learning rates? Have you conducted sensitivity analysis to understand the impact of these hyper-parameters on the performance?
>
> To better understand the properties of our proposed CPTPP, we conduct hyper-parameter studies on an important term, the dimension size of the personalized prompt. By fixing all the other hyper-parameters, we comprehensively examine the performance of three versions of the proposed CPTPP on all the datasets with different prompts. Specifically, the size of the personalized prompt is selected from $\{8, 16, 32, 64, 128, 256\}$. We choose two metrics, Precision@5 and NDCG@5, to demonstrate CPTPP's performance variations with different prompt sizes. All the experiment results are shown in Figure 3 and Figure 5 in the Appendix.
>
> (i) The first thing we can observe is that, in most cases, CPTPP has the best performance when the prompt size is not larger than the dimensionality of user embeddings. A potential reason is that sizeable prompt dimensions would introduce more noise into pre-trained user embeddings, disturbing the structural semantics extracted from the user-item interaction graph by graph contrastive learning.
> (ii) We also notice a significant performance improvement when prompt size is 256 in several cases, such as CPTPP-M on dataset ML-1M and CPTPP-R on dataset Gowalla.
> However, they still fail to significantly outperform the CPTPP model, which has a much smaller prompt size. Therefore, small prompt size for prompt-tuning is a better option in practice as they achieve a relatively good recommendation quality and higher efficiency.
>
> Q2. How generalizable are the prompt engineering techniques used in this study? Can they be applied to other recommendation tasks or different prompt-based methods beyond the proposed approach?
>
> The proposed method can be applied to various graph-based recommendation tasks and enhance the recommendation performance when no side information is available. Please note that the final outputs of our proposed method are the user embeddings enriched by the personalised prompts and item embeddings. The acquired user and item representations can be fed into any downstream tasks that accept such inputs. Therefore, the generalizability of our proposed method is promising.
>
> L1. Could you discuss potential challenges and limitations of prompt-tuning in recommendation systems? Are there any scenarios where prompt-tuning might not be suitable or effective?
>
> The challenges of applying prompt learning are time-consuming and require experts knowledge to design the prompts. Though a novel paradigm, soft prompt, is proposed, it requires auxiliary information like user profiles and extra computation resources to generate prompts according to the auxiliary information. The prompt-tuning paradigm may not work when the computation source is limited (e.g., edge device). Moreover, the prompt-tuning is tailored for pre-trained models. If there is no pre-trained models get involved, the prompt-tuning cannot be applied.

---

> > ### Comment · Reviewer_1hDm · 2023-08-13
> >
> > I appreciate the response from the authors, which has addressed most of my previous conerns. Generally, I think the topic studied in this paper is timely and interesting, while some details can be improved. I would like to update my scores accordingly.

---

### Official Review · Reviewer_NDuk · 2023-07-04

**Soundness:** 2 fair
**Presentation:** 2 fair
**Contribution:** 2 fair
**Rating:** 4
**Confidence:** 3

**Summary:**

This paper proposes a prompt-enhanced framework for GCL-based recommender named CPTPP. At the core of CPTPP is a personalized user prompts generation framework that summarizes user profiles in graph recommender systems. The generated user prompts are then integrated with pre-trained user embeddings when applied in downstream tasks. Empirical results on three benchmark datasets shows that CPTPP is able to outperform state-of-the-art baseline alternatives such as SimGCL.

**Strengths:**

- CPTPP achieves strong empirical performance, outperforming strong baseline approaches such as SimGCL.
- Graph contrastive pre-training is useful in the recommendation scenario with many potential downstream applications.
- Code is available which improves reproducibility.

**Weaknesses:**

- CPTPP relies on exploiting historical interaction records, adjacency matrix, and high-order user relations for generating personalized user prompts. The personalized user prompts can thus be viewed as features engineered from user-item interactions. When incorporating feature engineering into the model, it is not surprising that there will be an improvement in terms of performance.
- Performance is not much better even with the auxiliary information used.

**Questions:**

Why does combining personalized user prompts with pre-trained user embedding help narrow the distinct targets between pre-training and downstream tasks? The paper explains in detail how the personalized user prompts are created, but does not explain why this results in narrowed distinction.

**Limitations:**

CPTPP relies on mining historical interaction records, adjacency matrix, and high-order user relations for generating personalized user prompts, and therefore it is possible that the improved performance primarily comes from the extra features used.

---

> ### Author Rebuttal · Authors · 2023-08-09
>
> Thanks for your valuable comments and questions. We will revise our paper according to them. Now, we would like to respond to your suggestions and questions.
>
> W1. CPTPP relies on exploiting historical interaction records, adjacency matrix, and high-order user relations for generating personalized user prompts. The personalized user prompts can thus be viewed as features engineered from user-item interactions. When incorporating feature engineering into the model, it is not surprising that there will be an improvement in terms of performance.
>
> Our method is inspired by a novel paradigm, soft prompt. The original goal of prompt learning is to elicit the pre-trained model. However, it is time-consuming and requires experts while designing hard prompts. The soft prompt paradigm addresses such limitation by utilising side information like user profiles to adaptively generate soft prompts. The generated prompts help the proposed method achieve better performance. Please notice that, the baselines also process the user-item interaction graphs but fail to outperform our method, which shows the advantages of our method to process the interaction graph, which is soft prompting.
>
> W2. Performance is not much better even with the auxiliary information used.
>
> Please note that there is no auxiliary information available. The settings of our research is graph-based recommendation without side information. Only user-item interaction graph is available for each method. We first propose acquiring various user profiles based on user-item interaction graph and generating soft prompts based the acquired user profiles. No side information used in our method.
>
> Q1. Why does combining personalized user prompts with pre-trained user embedding help narrow the distinct targets between pre-training and downstream tasks? The paper explains in detail how the personalized user prompts are created, but does not explain why this results in narrowed distinction.
>
> One of the advantages of prompt learning is to narrow the distinct targets between pre-training and downstream tasks, which is verified by many research works in the community of prompt learning. If the reviewer wants to know more about the mechanism and advantages of prompt learning, please refer to the section of related work, where we list some critical literature which is the basement of prompt learning.
>
> L1. CPTPP relies on mining historical interaction records, adjacency matrix, and high-order user relations for generating personalized user prompts, and therefore it is possible that the improved performance primarily comes from the extra features used.
>
> Thanks for acknowledging that our proposed personalised prompts improve the performance. Please note that there is no extra features used in our method, the personalised prompts are generated solely based on the user-item interaction graph.

---

> > ### Comment · Reviewer_NDuk · 2023-08-17
> >
> > I would like to thank the authors for the detailed response to my questions and concerns. The authors' rebuttal has addressed most of my concerns, although I still have concern about the limited performance improvement. I have therefore raised my score accordingly.

---

### Official Review · Reviewer_UMuQ · 2023-07-07

**Soundness:** 2 fair
**Presentation:** 3 good
**Contribution:** 2 fair
**Rating:** 4
**Confidence:** 4

**Summary:**

This paper proposes a prompt-tuning approach for GCL-based recommender systems. A framework consisting of a GCL module, a prompt generation module and a recommendation module is developed. Both ablation study and hyper-parameter study are conducted.

**Strengths:**

- This paper studies an interesting research problem, prompt tuning for GCL-based recommendation, which gets rid of a combination of two quite different targets.
- The paper is easy to follow.
- Both ablation study and hyper-parameter study are conducted.

**Weaknesses:**

- The improvements over existing baselines are not significant. For example, SimGCL outperforms the proposed CPTPP approach on Gowalla dataset w.r.t NDCG@20.
- The visualization of Figure 2 is not convincing to me.
- There is no doc or README in the released codes.
- The authors did not provide the reasons for selecting the adopted baselines.
- What is the computational complexity of the proposed method compared with existing approaches?

**Questions:**

See the weakness.

---

> ### Author Rebuttal · Authors · 2023-08-09
>
> Thanks for your review. We would like to respond to your comments and questions in the following.
>
> W1. The improvements over existing baselines are not significant. For example, SimGCL outperforms the proposed CPTPP approach on Gowalla dataset w.r.t NDCG@20.
>
> Thanks for your conscientious and detailed review. We will keep tuning the proposed model. Currently, CPPTPP outperforms the baselines in most cases and has a very close performance compared to SimGCL on dataset Gowalla regarding the metric NDCG@20.
>
> W2. The visualization of Figure 2 is not convincing to me.
>
> The reviewer PJh3 has the same concern regarding the visualisation results. Please refer to Q3 in the reponse to reviewer PJh3.
>
> W3. There is no doc or README in the released codes.
>
> Thanks for your reminder. We will add the related document later. The document is also available in the SelfRec project on Github as I officially stated that our implementation is based on the source codes of SelfRec.
>
> W4. The authors did not provide the reasons for selecting the adopted baselines.
>
> There are three types in the baselines we selected. BPR-MF is a conventional recommendation method. BUIR and SelfCF are both contrastive learning-based recommendation systems. NCL and SimGCL are two representative GCL-based recommendation systems. As to our proposed method, we take SGL as the backbone, the predecessor version of SimGCL. We will add related information in the main content later.
>
> W5. What is the computational complexity of the proposed method compared with existing approaches?
>
> Our proposed method is a framework to address the limitations of current GCL-based recommendation methods. The complexity of the proposed methods depends on the complexity of backbone GCL method and the user profile generation method. Therefore, there is no fixed complexity for our proposed method.

---

### Official Review · Reviewer_PJh3 · 2023-07-07

**Soundness:** 3 good
**Presentation:** 3 good
**Contribution:** 3 good
**Rating:** 5
**Confidence:** 4

**Summary:**

This paper proposes a prompt-enhanced framework for GCL-based recommender system, called CPTPP. CPTPP reforms the existing GCL-based recommendation methods with the prompt tuning mechanism to fully exploit the advantages of GCL in the pre-training phase instead of combining the contrastive loss with downstream objectives. The authors summarize three user profiles derived from the user-item interaction graph as the inputs, without requiring extra side information, for the prompt generator. Extensive experiments on real-world datasets have demonstrated the effectiveness.

**Strengths:**

1. The proposed CPTPP reforms the existing GCL-based recommendation methods by separating the GCL pre-training and the downstream recommendation task using the prompt tuning mechanism. The idea is innovative for GCL based recommendation.

2. Integrating prompts could better elicit the knowledge within the pre-trained user and item embeddings. The authors propose three different prompt generation methods, which can be applied to situations where users’ side information is not available.

3. The writing is good, which is well organized and easy to read. Most experimental details are provided.


**Weaknesses:**

1. Generating prompts for users in recommender systems has been proposed in existing work. The difference mainly lies in that this paper addresses the GCL-based recommendation situation, where no side information of users is available.

2. As an empirical study, some important details are not fully explained. For example, in section 2.2, the authors stated that “we can adopt various GCL learning methods …, to obtain high-quality user and item embeddings.”. Which GCL method did the authors actually adopt in their experiments? This is not claimed in the paper.

I have some other questions about the experiments. Please refer to the “Questions” section.

3. The Ablation Studies (section 3.2.3) actually provides horizontal comparisons among CPTPP-M, CPTPP-H, and CPTPP-R, instead of standard ablation studies. It is nice to provide some insights into the experiments. But I wonder to what extent the proposed methods improve the performance compared with models without the personalized prompts.


**Questions:**

1. What are the backbone GCLs of these models in the experiments? Do the generation modules perform differently on different GCLs?

2. The original goal of prompt designing is to better elicit the knowledge contained in the pre-trained model for downstream applications. The second proposed method (CPTPP-M) for prompt generation actually obtains user and item embedding using adjacency matrix factorization separately which is irrelevant from GCL. Moreover, CPTPP-M performs the best half of the time. Can the authors explain the reason?

3. Section 3.2.1, “As suggested in [15], the more uniform the embedding distribution is, the more capability to model the diverse preferences of users the method has”. Actually, this is a “speculation” in the paper [15] instead of a conclusion. Moreover, “uniform” distribution is quite ill-defined here. I find it hard to differentiate which one of the figures is more uniformly distributed. Are there any quantitative measurement methods?

4. It seems that the proposed methods are not limited to user embeddings. Can the personalized prompt generation methods be symmetrically applied to items?

---

> ### Author Rebuttal · Authors · 2023-08-09
>
> Thanks for recognising the contribution and novelty of our research work. We appreciate your suggestions and questions, and we will revise the manuscript according to your comments. The following is our response to your comments.
>
> W1. Generating prompts for users in recommender systems has been proposed in existing work. The difference mainly lies in that this paper addresses the GCL-based recommendation situation, where no side information of users is available.
>
> We totally agree with you. Generating prompts in conventional recommender systems is a widely explored topic, focusing on utilising various side information of users and items. However, in our settings, we focus on the graph-based recommender system where only the user-item interaction graph is available, and no side information is provided. And the scope of our work is to adaptively construct prompts from the interaction graph to enhance the recommendation performance.
>
> W2. As an empirical study, some important details are not fully explained. For example, in section 2.2, the authors stated that “we can adopt various GCL learning methods …, to obtain high-quality user and item embeddings.”. Which GCL method did the authors actually adopt in their experiments? This is not claimed in the paper.
>
> We apologise for neglecting the details about the base GCL model we adopt. To ensure that delicate GCL models do not cause performance improvement, we take SGL, a vanilla GCL model without sophisticated components, as the base model. SGL is the predecessor of the baseline SimGCL. We also strictly follow the evaluation protocols in GraphCL and InfoGraph to ensure a fair comparison. We will add related descriptions in the manuscript later. Thanks for your reminder.
>
> W3. The Ablation Studies (section 3.2.3) actually provides horizontal comparisons among CPTPP-M, CPTPP-H, and CPTPP-R, instead of standard ablation studies. It is nice to provide some insights into the experiments. But I wonder to what extent the proposed methods improve the performance compared with models without the personalized prompts.
>
> Thanks for your suggestions. It is important to show the improvement brought by personalised prompts explicitly. Considering the base model we adopt is SGL, we will conduct experiments on SGL to compare it with our proposed method to verify the advantages of personalised prompts generation.
>
> Q1. There are three types in the baselines we selected. BPR-MF is a conventional recommendation method. BUIR and SelfCF are both contrastive learning-based recommendation systems. NCL and SimGCL are two representative GCL-based recommendation systems. As to our proposed method, we take SGL as the backbone, the predecessor version of SimGCL.
>
> The generation modules in our proposed method have the same operations as our framework and can take various GCL-based recommendation methods as the backbone. The generation modules are independent of the GCL module. We can combine the generated prompts with the outputs of the GCL module to help to improve the recommendation performance.
>
> Q2. We agree that the original goal of prompt learning is to elicit the pre-trained model. However, it is time-consuming and requires experts while designing prompts. There are research works proposed a soft prompt paradigm [1][2] to address such limitations. Such a novel paradigm inspires our work. The fundamental procedure of soft prompts is to adaptively generate prompts based on side information like user profiles.
>
> Three user profiles are proposed in our work for graph-based recommendation scenarios without side information. All of them, including adjacency matrix factorisation (MF), are highly related to recommendation tasks. MF is a conventional recommendation method. It can produce high-quality low-dimension embeddings for users and items, reflecting the preferences of users and item features. So we can utilise the outputs of MF as the user profile for the downstream personalised prompt generation. Please note that the user profile acquisition is optional to be highly related to GCL.
>
> Historical interaction records and high-order user relations are simple aggregation methods, combining user embeddings and items related to the target users. However, adjacency matrix factorisation is a machine learning method that can help embed initial user and item representations to a low-dimension latent space. A trainable embedding method can produce better user profiles than the two simple aggregation methods. That is a potential reason why CPTPP-M has better performance.
>
> [1] Taylor Shin, Yasaman Razeghi, Robert L. Logan IV, Eric Wallace, and Sameer Singh. Autoprompt: Eliciting knowledge from language models with automatically generated prompts.
> [2] Yiqing Wu, Ruobing Xie, Yongchun Zhu, Fuzhen Zhuang, Xu Zhang, Leyu Lin, and Qing He. Personalized prompts for sequential recommendation.
>
> Q3. Thanks for your reminder. We acknowledge the limitations of our visualisations. Determining which is more uniformly distributed according to the visualisation results is not straightforward. There is a potential alternative measurement. We can first generate a 2-dimension uniform distribution. Then, we use metrics like Kullback-Leibler divergence and Wasserstein distance to measure the difference between the generated uniform distribution and the distribution of obtained user embeddings. A slight divergence or distance indicates a minor difference. In this way, we may offer a quantitative method to measure the "uniformity". We will add quantitative results together with the visualisation to demonstrate the quality of obtained user embeddings.
>
> Q4. Yes, it can be symmetrically applied to items. Should the participants in some scenarios request to recommend users for a specific item to construct the list of potential clients, they can do it symmetrically.

---

> > ### Comment · Reviewer_PJh3 · 2023-08-19
> > **Response to Rubttal**
> >
> > I appreciate the authors for their efforts in addressing my concerns. My questions have been addressed. Generally, this is a satisfactory paper, and the experiments could be further improved as we discussed. I am still positive about this paper.

---

### Comment · Area_Chair_DfKQ · 2023-08-13

Dear reviewers, please take your time to carefully review the author's rebuttal and provide your response. Thank you!

---

### Decision · Program_Chairs · 2023-09-21

**Decision:**

Accept (poster)

**Comment:**

Overall, the reviewers all believe that this work attempts to address an important research problem and the proposed method has sufficient novelty. Although some reviewers raised questions about the adequacy of the experiments and the clarity of the description, the consensus in the discussion is that the authors would be able to address these issues well in their final version.